# Recapitulation of Peste des Petits Ruminants (PPR) Prevalence in Small Ruminant Populations of Pakistan from 2004 to 2023: A Systematic Review and Meta-Analysis

**DOI:** 10.3390/vetsci11060280

**Published:** 2024-06-19

**Authors:** Saad Zafar, Muhammad Shehroz Sarfraz, Sultan Ali, Laiba Saeed, Muhammad Shahid Mahmood, Aman Ullah Khan, Muhammad Naveed Anwar

**Affiliations:** 1Institute of Microbiology, Faculty of Veterinary Science, University of Agriculture, Faisalabad 38000, Punjab, Pakistan; 2011ag2402@uaf.edu.pk (S.Z.); 2017ag8190@uaf.edu.pk (M.S.S.); sultanali@uaf.edu.pk (S.A.); shahiduaf@uaf.edu.pk (M.S.M.); 2Institute of Microbiology, Government College University, Faisalabad 38000, Punjab, Pakistan; 2017-gcuf-04740@gcuf.edu.pk; 3Department of Pathobiology, University of Veterinary and Animal Sciences (Jhang Campus), Lahore 54000, Punjab, Pakistan

**Keywords:** peste des petits ruminants, small ruminants, pooled prevalence, Punjab, Sindh, Baluchistan, KPK, PPR

## Abstract

**Simple Summary:**

Peste des petits ruminants (PPR), commonly referred to as goat plague or ovine rinderpest, is an infectious viral ailment that specifically targets wild or domesticated small ruminants, including sheep and goats. The study examined the information gathered from several sources such as research publications to evaluate the overall impact of PPR in Pakistan. The overall pooled prevalence in Pakistan was calculated to be 51%. Among various regions, the level of pooled prevalence of PPR presented a non-significant difference and was almost the same. Moreover, the investigation found other factors that influence the spread of PPR, including animal husbandry techniques, immunization, and geographical factors. In summary, the review of the literature provides significant information about the occurrence and distribution of PPR in Pakistan. There is a dire need to implement efficient management measures to reduce the negative effects of this disease on small ruminant populations.

**Abstract:**

Peste des petits ruminants (PPR) is an extremely transmissible viral disease caused by the PPR virus that impacts domestic small ruminants, namely sheep and goats. This study aimed to employ a methodical approach to evaluate the regional occurrence of PPR in small ruminants in Pakistan and the contributing factors that influence its prevalence. A thorough search was performed in various databases to identify published research articles between January 2004 and August 2023 on PPR in small ruminants in Pakistan. Articles were chosen based on specific inclusion and exclusion criteria. A total of 25 articles were selected from 1275 studies gathered from different databases. The overall pooled prevalence in Pakistan was calculated to be 51% (95% CI: 42–60), with heterogeneity *I*^2^ = 100%, *τ*2 = 0.0495, and *p* = 0. The data were summarized based on the division into five regions: Punjab, Baluchistan, KPK, Sindh, and GB and AJK. Among these, the pooled prevalence of PPR in Sindh was 61% (95% CI: 46–75), *I*^2^ = 100%, *τ*2 = 0.0485, and *p* = 0, while in KPK, it was 44% (95% CI: 26–63), *I*^2^ = 99%, *τ*2 = 0.0506, and *p* < 0.01. However, the prevalence of PPR in Baluchistan and Punjab was almost the same. Raising awareness, proper surveillance, and application of appropriate quarantine measures interprovincially and across borders must be maintained to contain the disease.

## 1. Introduction

The livestock sector in Pakistan is an essential component of the country’s economy, contributing about 62.68% to the agricultural industry and 14.36% to the national gross domestic product (GDP). Each household typically maintains a flock of five to six sheep or goats. In Pakistan, the population of goats is 84.7 million, while the number of sheep is 32.3 million. Small ruminants are often reared by individuals of low socioeconomic status and farmers in Pakistan as a supplementary livelihood; however, much attention is now being paid to the commercial-level farming of sheep and goats. In addition, a significant amount of farmers’ income in Pakistan’s mountainous regions comes from rearing sheep and goat herds [1,2]. The disease in question poses a significant danger to the sustainable livelihoods of farmers and the food security of regions spanning Asia, the Middle East, and Africa [3,4].

Peste des petits ruminants (PPR), sometimes referred to as ‘ovine rinderpest’, ‘Kata’, ‘stomatitis–pneumonitis syndrome’ [5], and ‘goat plague’ [6], is a viral disease that is highly contagious and mainly affects sheep and goats [7,8]. The highly infectious nature of this disease leads to its classification as a transboundary animal disease by the World Organization for Animal Health (OIE) [9,10]. The etiological agent of the disease is the peste des petits ruminants virus (PPRV), which belongs to the genus Morbillivirus within the family *Paramyxoviridae* [11,12]. The PPRV genome consists of 15,948 nucleotides and is a negative-sense, single-stranded RNA structure that is non-segmented [13,14].

PPRV has a single serotype and is genetically classified into four different lineages (I, II, III, and IV) based on partial sequence analysis of the fusion (F) and nucleoprotein (N) genes. Lineages I and II are specifically isolated from Western Africa [15,16]. Lineage III has a restricted distribution, mainly including East Africa and Arabia, with a single instance of the virus associated with this lineage being identified in Southern India [17]. The emergence of new viruses is associated with Lineage IV, a novel lineage widely distributed throughout several Asian countries and some Northern and Middle African countries [18]. The PPRV lineages provide insights into the epidemiological aspects of the disease by explaining the origin of the viral strain responsible for initiating a new epidemic [19]. PPR virus can decimate populations of immunocompromised hosts, giving rise to outbreaks that can adversely impact a country’s economy and compromise food security and farmers’ livelihoods [20].

PPR also has significant rates of morbidity, reaching up to 100%, and mortality up to 50–80% among vulnerable animals [5,21]. In Pakistan, the incidence of disease results in substantial economic losses of up to Rs 20.5 billion per annum [21]. The transmission of PPRV often necessitates direct contact with animals that are affected. The primary transmission routes of PPRV are oral and aerosol among cohabitating animals; oral and oculonasal discharge are the primary sources of infection [22,23]. PPR disease may manifest in acute or sub-acute forms. Clinically acute disease is associated with an elevated fever (up to 41.1 °C, which may last for a week), depression, anorexia, and a dry muzzle. The sub-acute form may present the same signs and symptoms in a milder form for a longer duration. Serous nasal and ocular excretions develop gradually into a mucopurulent discharge, which may block the nares and encrust the muzzle, causing the animal to snort and sneeze, whereas the ocular discharges may mat the eyelids together. Watery bloody diarrhea is common in the later stages of the disease. Coughing, pneumonia, pleural rales, and abdominal breathing are also present, and death occurs within one week of the onset of illness [24]. 

This disease was previously believed to be mostly confined to the West African region. However, current knowledge suggests that its dissemination has occurred across a vast geographical area, spanning a significant portion of West, Central, and East Africa, with its presence extending towards Southern and Western Asia [3]. However, the variations in the prevalence and intensity of PPR outbreaks may be associated with differences in herd management practices in different geographical locations, the geography of these regions, and other relevant aspects. The disease shows endemicity in several geographical locations, including Morocco, Sub-Saharan Africa, the Arabian Peninsula, Turkey, the Middle East, Iraq, Iran, Pakistan, Bangladesh, India, Nepal, Tibet, China, Tajikistan, and Kazakhstan [25]. In Pakistan, PPR was initially documented in 1991 in Punjab province [21]. However, detailed information on the epidemiology and incidence of the PPR virus is limited in various parts of the country. The current study aims to evaluate the currently available data regarding the prevalence of PPR in different districts of Pakistan to offer a clear understanding of the disease burden and epidemiology.

## 2. Materials and Methods

### 2.1. Review Assessment Protocol

The present review was in accordance with guidelines for systematic reviews available on Preferred Reporting Items for Systematic Reviews and Meta-Analyses (PRISMA) [26]. The review was conducted in the following steps, including (1) conducting searches across the database for potentially relevant articles, (2) determining the suitability of the articles for inclusion in the review, (3) assessing the relevance of the selected articles, and (4) carrying out the extraction, screening, and analysis of data. The above protocol was well described before commencing the systematic review and meta-analysis. 

### 2.2. Searching Methodology

A comprehensive approach was adopted to find original research articles published between January 2004 and August 2023, and the seroprevalence and distribution pattern of PPR was examined in Pakistan. We explored various electronic web database systems for a comprehensive literature search, including Google Scholar, Research Gate, PubMed, ScienceDirect, Scinapse, Web of Science, and Crossref. All the databases were accessed for the last time in August 2023. The searched terms included “Pakistan”, “Peste des petits ruminants”, “PPR”, “Punjab”, “Sindh”, “Khyber Pakhtunkhwa”, “Baluchistan”, “Azad Jammu and Kashmir”, “Gilgit–Baltistan”, “prevalence”, “incidence”, “frequency”, “characterization”, “detection”, “identification”, “isolation c-ELISA”, “Ic-ELISA”, “PCR”, “small ruminants”, “sheep”, and “goat”. By considering the study’s objective, various Boolean keywords were devised, including AND (same category word) and OR (within category word).

Furthermore, the list of the selected articles was provided for further assessment to maximize the chance of obtaining other articles. During the search, we acquired all the articles directly from the journals in which they were published. The analysis includes all the articles that met the inclusion criteria. 

### 2.3. Inclusion and Exclusion Criteria

All the obtained articles were confirmed and cross-checked before their inclusion in the meta-analysis and systemic review. Articles that fulfilled the given criteria were selected for the use of the underlying studies. 

(1)Articles focused on peste des petits ruminants in small ruminants (sheep and goats) were incorporated.(2)Articles with clear information about the sample size, total positive number, and detection method were included.(3)Peer-reviewed articles that were published between January 2004 and August 2023 were included.(4)All the articles containing data on PPR from various regions of Pakistan were included in the study.

The final list of the articles was compiled based on the abovementioned criteria. Articles were excluded from consideration based on the following criteria.

(1)Duplicate articles that had been peer-reviewed previously and dismissed.(2)Articles that were not per the established inclusion criteria.(3)Articles that did not match the scope of our study.(4)Articles unavailable in full text or containing only a title and/or abstract.(5)Articles were excluded if they were unpublished, under-review articles or meta-analyses, or were conference/symposium or meeting abstracts.

### 2.4. Data Extraction

Two authors, Saad Zafar and Muhammad Shehroz Sarfraz, independently extracted the data. After thoroughly reviewing the complete texts of the studies, they systematically assessed the eligibility criteria for the studies and collected appropriate data, which were then entered into a spreadsheet. Subsequently, the other reviewers, Aman Ullah Khan, Sultan Ali, and Naveed Anwar, cross-verified the gathered data. The data comprised various elements such as the study’s title, name of the first author, the publication year (2004–2023), geographical location (province) of the study, duration of the study, the categories and types of samples, sample size, and the prevalence of PPR with the detection method (serological or molecular). 

### 2.5. Data Analysis

All the collected data were initially imported to an Excel sheet (Microsoft/Office 365, Redmond, WA, USA) for data analysis. After arranging the data, it was exported to R (version 4.3.2, The R Foundation for Statistical Computing, Vienna, Austria). The meta-analysis was conducted using RStudio (v2023.03.0-daily+82.pro2) by employing the metaprop codes available within the meta package (v 6.5-0) [27] of the R program. To enumerate the pooled prevalence along with a 95% confidence interval (CI) of PPR in small ruminants in Pakistan, we employed the maximum likelihood approach within the random effects model. Cochran’s Q test and the inconsistency index (*I*^2^) were used to examine the relevance of statistical heterogeneity among the studies. In this current analysis, a *p*-value of less than 0.05 and an *I*^2^ value of more than 75% were signs of significant statistical variance [28] by using the “forest” code in the meta package, and “jpeg” and “dev. off()” codes. Forest plots were prepared, and images were extracted from the “R” plot. Additionally, we used RStudio to compute the prevalence and corresponding 95% confidence intervals (Cis) for PPR in small ruminants in Pakistan using the Wilson/Brown hybrid method [29].

The authors SZ and MSS performed all the analyses.

## 3. Results

### 3.1. Study Inclusion Details and Quality Assessment of Biases

Per the PRISMA guidelines, 1275 articles (details for these articles such as author name and published year are provided in Appendix A) were considered for the initial screening. The selection of 396 articles to be assessed as eligible was made after removing duplicates and articles that did not follow the review concept or were lacking in detail. Finally, the eligibility criteria for our current systematic analysis and meta-analysis were successfully met by 25 scientific articles, and these were further subjected to quality assessment of biases, as shown in Figure 1. The studies covered in the selected publications were carried out between 2002 and 2021, while the articles were published between 2004 and 2023. The number of studies published in each year are depicted in Figure 2. The published studies were conducted either in a single district or a cluster of districts in Pakistan, and we compiled them province-wise in Table 1.

#### 3.1.1. Publication Bias

Publication bias poses a significant problem in systematic reviews and meta-analyses, affecting the validity and interpretation of findings [55]. This study used funnel plot-based methodologies such as a visual inspection of a funnel plot, regression analysis, and rank tests to assess publication bias. A funnel was plotted, with the arcsine-transformed proportion plotted along the X-axis, while the standard error was plotted along the Y-axis. The key reason for integrating the arcsine-based transformation in our analysis was its substantial advantage in terms of variance stabilization [55]. In Figure 3, most of the studies appear scattered, while a handful are positioned within the funnel, illustrating the publication bias. To address the issue of publication bias, we employed a meta-regression analysis, utilizing sample size as the indicator for assessing bias factor. The results indicated non-significance (*p* > 0.05), effectively negating the impact of publication bias on the study.

#### 3.1.2. Meta-Regression for the Identification of the Factors Affecting the Heterogeneity 

Univariate meta-regression was used to discover possible factors that might affect the size and direction of the overall estimate of heterogeneity. The result of the meta-regression in Figure 4 reveals that the use of detection methods had a notable impact on the total diversity, at a significance level of 5%. The variables, such as the type of test, species, sample sizes, and the year, exhibited statistical significance. The estimated findings emphasize the need for subgroup analysis and sensitivity analysis to enhance and deepen our understanding of the prevalence rates of PPR. The size of the bubble shown in the bubble plot for univariate meta-regression in Figure 4 indicates the respective study’s weight, meaning a large bubble indicates a larger sample size and a small bubble shows a smaller sample size. 

### 3.2. Methods for Detection of PPR

Molecular and serological assays were employed for the detection of PPR. Only four out of twenty-five studies used the molecular assay, and the remaining twenty-one used the serological method to detect PPR. Moreover, only one study under the serological assay category used the agar gel immunodiffusion (AGID) test for detection, and two employed the hemagglutination inhibition (HI) test. Of twenty-one studies, sixteen employed c-ELISA to detect antibodies against PPRV, and only three employed Ic-ELISA to detect antigens (Table 1). The effect of the detection method is presented by the correlation plot in Figure 5. The variation in correlation indices in the plot signifies that the choice of detection method during the estimation of proportion may affect the prevalence rates.

### 3.3. Region and Animal Species Reported

A total of 25 articles covering the different provinces and regions of Pakistan (Punjab, Sindh, Khyber Pakhtunkhwa, Baluchistan, Gilgit–Baltistan, Azad Jammu and Kashmir, and Islamabad Capital Territory) were included in this study. The number of articles published from Punjab province was seven, Sindh six, Khyber Pakhtunkhwa three, Gilgit–Baltistan and Azad Jammu and Kashmir two, and Islamabad Capital Territory one, and the data from six articles were under the title “Pakistan”. No specific study was available for Baluchistan province, so we extracted the data from the remaining six articles covering the overall PPR prevalence estimate in Pakistan. Since there was only one study from Islamabad Capital Territory, we included that in Punjab, as it shares the boundary with Rawalpindi, a city of Punjab province. Only two species of domesticated small ruminants were targeted: sheep and goats.

### 3.4. Prevalence Estimates

The meta-analysis employing random effects for sheep and goats revealed that the pooled prevalence of PPR was 51% (95% CI: 42–60), with heterogeneity *I*^2^ = 100%, *τ*2 = 0.0495, and *p* = 0 (Figure 6). 

The data from all over Pakistan presented 10,995 samples identified as positive from a total of 28,020 samples. The overall pooled prevalence was estimated to be 51% (CI: 42–60%), with a minimum value of 18% (CI: 17–19%) and a maximum value of 100% (CI: 83–100%). The data illustrated heterogeneity, *I*^2^ = 100%, and *p*-value = 0. The data for province-wised pooled prevalence are illustrated in Figure 7.

The data from Sindh province show that 5868 samples were positive out of 15,184 samples. The overall pooled prevalence was estimated to be 61% (CI: 46–75%), with a minimum value of 28% (CI: 27–29%) and a maximum value of 98% (CI: 97–99%). The data demonstrated heterogeneity, *I*^2^ = 100%, and *p*-value = 0.

The data from Punjab province show that 2694 samples were identified as positive out of 5186 samples. The overall pooled prevalence was estimated to be 54% (CI: 42–65%), with a minimum value of 22% (CI: 17–28%) and a maximum value of 100% (CI: 83–100%). The data demonstrated heterogeneity, *I*^2^ = 98%, and *p*-value < 0.01.

The data from Baluchistan province show that 168 samples were identified as positive out of 337. The overall pooled prevalence was estimated to be 53% (CI: 41–65%), with a minimum value of 45% (CI: 38–52%) and a maximum value of 65% (CI: 51–78%). The data demonstrated heterogeneity, *I*^2^ = 76%, and *p*-value = 0.01.

The data from the GB and AJK regions show that 885 samples were positive out of 1842 samples. The overall pooled prevalence was estimated to be 51% (CI: 32–71%), with a minimum value of 27% (CI: 12–46%) and a maximum value of 84% (CI: 78–88%). The data demonstrated heterogeneity, *I*^2^ = 98%, and *p*-value < 0.01.

The data from Khyber Pakhtunkhwa show that 1245 samples were positive out of 5363. The overall pooled prevalence was estimated to be 44% (CI: 26–63%), with a minimum value of 18% (CI: 17–19%) and a maximum value of 83% (CI: 76–88%). The data demonstrated heterogeneity, *I*^2^ = 99%, and *p*-value < 0.01.

Pearson’s chi-squared test applied to the gathered data for prevalence in sheep and goats revealed that there was no significant difference between the prevalence rates of PPR in the respective species, as shown in Table 2. The *p*-value of 0.9999 signifies that both groups are independent in terms of being affected by PPR.

## 4. Discussion

Peste des petits ruminants (PPR), an infectious viral disease with high transmissibility that affects domesticated and wild small ruminants, has emerged as a significant problem in the livestock sector in Pakistan [52]. The objective of this meta-analysis was to consolidate and assess the currently available research on PPR in Pakistan to provide a comprehensive understanding of the disease’s epidemiology, prevalence, and control strategies. The occurrence of PPR has been documented in several regions in Pakistan. However, the meta-analysis included 25 papers out of the initial pool of 48 acceptable articles due to inter-rater disagreements. 

An author previously conducted a meta-analysis in 2020 using the random effect model to evaluate the prevalence of PPR in small ruminants (sheep and goats) in 34 countries, including Pakistan. The findings indicated a combined prevalence rate of 43.55% (95% CI: 38.22–48.88) in Pakistan [56]. The present study analysis revealed the high prevalence of PPR in small ruminants all over Pakistan. The meta-analysis employing random effects for sheep and goats showed that the pooled prevalence of PPR was 51% (95% CI: 42–60) with heterogeneity, *I*^2^ = 100%, *τ*2 = 0.0495, and *p* = 0 (Figure 4). Moreover, the study findings within the specific geographic area showed that the prevalence of PPR in Sindh was 61% (95% CI: 46–75), *I*^2^ = 100%, *τ*2 = 0.0485, and *p* = 0, followed by Punjab with 54% (95% CI: 42–65), *I*^2^ = 98%, *τ*2 = 0.0461, and *p* < 0.01, Baluchistan with 53% (95% CI: 41–65), *I*^2^ = 76%, *τ*2 = 0.0083, and *p* = 0, Gilgit–Baltistan and Azad Jammu and Kashmir with 51% (95% CI: 32–71), *I*^2^ = 98%, *τ*2 = 0.0469, and *p* < 0.01, and Khyber Pakhtunkhwa 44% (95% CI: 26–63), *I*^2^ = 99%, *τ*2 = 0.0506, and *p* < 0.01 **(**Figure 7). The spatial distribution of PPRV in Pakistan for the estimation of PPR prevalence in sheep and goats is shown in Figure 8. Moreover, our study only focused on those publications that employed serological assays and molecular detection (PCR) for the confirmatory diagnosis of PPR in small ruminants. The Sindh province was shown to have the highest seroprevalence, which may be attributed to the significant number of sheep and goats in the region. The prevalence rates in the provinces of Punjab and Baluchistan were similar, perhaps because of similar risk circumstances, such as high population density and various migratory routes in these areas [50].

Our comprehensive research shows that PPR was consistently prevalent in Pakistan, indicating an endemic status throughout the years. The disease has affected both domesticated and wild populations of small ruminants along with unusual hosts like camels and buffaloes [12,57,58], resulting in significant economic losses and affecting farmers’ livelihoods, particularly those in rural regions that rely significantly on livestock [13]. PPR has regional variations in its frequency throughout Pakistan. Due to the distribution of PPR in Pakistan, it is necessary to implement appropriate control measures [59]. Possible factors contributing to the widespread occurrence of PPR in sheep and goats may include the inter-provincial movement of diseased animals without sufficient quarantine measures (especially during Eid al-Adha days) [60,61,62]. Temporal analysis reveals that PPR worldwide displays seasonal variations, with outbreaks often occurring throughout certain months of the year, and is a similar case in Pakistan [19,63,64]. The seasonal fluctuations may be impacted by climatic factors such as temperature and humidity, which can influence the virus’s ability to survive and spread [65,66]. Comprehending these seasonal trends is essential for prompt intervention and immunization programs to alleviate the consequences of PPR [67]. The studies included in the present meta-analysis highlight many risk variables contributing to the dissemination of PPR in Pakistan. These factors involve migrating animals with disease, grazing in open pastures, insufficient knowledge among farmers about disease control, and inadequate vaccination rates, especially in distant and economically challenged regions (data available in Appendix A) [68]. The lack of regulation in the animal trade and the absence of quarantine procedures have also made it easier for diseases to spread within national and across international borders. The high prevalence status of the disease in Pakistan’s neighboring countries, as depicted in Figure 9, further hinders the control of the disease due to its transboundary nature. 

It is crucial to tackle these risk factors to achieve efficient disease control. Immunization against PPR has shown the potential to decrease the occurrence of the disease in Pakistan [54]. Multiple papers cited in this meta-analysis emphasize the efficacy of vaccination efforts in suppressing PPR outbreaks. In addition, there is a need to enhance the extent and uniformity of immunization programs, particularly in regions with a high risk of disease. Ensuring the availability of top-notch vaccines and bolstering veterinary services are essential to managing PPR [69]. Moreover, many areas of Pakistan may not have sufficient representation in the existing literature, restricting the applicability of our results. Furthermore, some papers included in this meta-analysis used serum samples to evaluate the prevalence of PPR, which may create bias in the study due to the presence of PPRV antibodies developed previously by vaccination or exposure to disease, as antibodies of PPRV can reportedly stay in the body for up to 12 months [70]. Eleven articles stated that samples were taken from non-vaccinated animals [29,30,33,34,36,37,38,45,46,49,50] and eleven articles [31,32,35,39,40,42,43,47,48,51,53] did not provide any data about vaccination status. The remaining three articles [41,44,52] mentioned that the vaccination history was recorded but did not provide any data in the texts. 

Although there are biological differences between sheep and goats, the variations in prevalence can be explained by factors such as how the samples were collected, how many animals there were in a particular area, management practices, and the specific type of virus lineage [21,71,72]. In addition, it is conceivable that PPRV preferentially infects goats over sheep or vice versa, depending on the endemic scenario. Furthermore, the severity of the disease may also vary from species to species [72]. The variations in the prevalence statistics provided in this study depend on variables such as the source of the sample, the technique used for detection, and the time frame of the investigation.

## 5. Conclusions

Peste des petits ruminants remains a substantial threat to Pakistan’s livestock sector. This meta-analysis highlights the need to enhance monitoring surveillance, improve vaccination coverage, and address the risk factors linked to disease transmission. Effective implementation of control measures and reducing the burden of PPR in Pakistan need collaboration among government agencies, veterinary authorities, and foreign organizations. Future studies should give priority to specific regions, improving the quality of data, and evaluating the effect of vaccination efforts on decreasing the incidence of PPR. These endeavors will aid in the development of evidence-based strategies for the control and ultimate elimination of PPR in Pakistan.

## 6. Limitation

It is important to recognize certain constraints of this meta-analysis. The variability in data quality and consistency among research may have potentially brought bias into the findings. Furthermore, it is vital to acknowledge the potential existence of publication bias, where research that provides favorable results may have a higher likelihood of being published. Moreover, heterogeneity in research, characterized by variations in methodologies and definitions of variables, presented difficulties in data synthesis.

## Figures and Tables

**Figure 1 vetsci-11-00280-f001:**
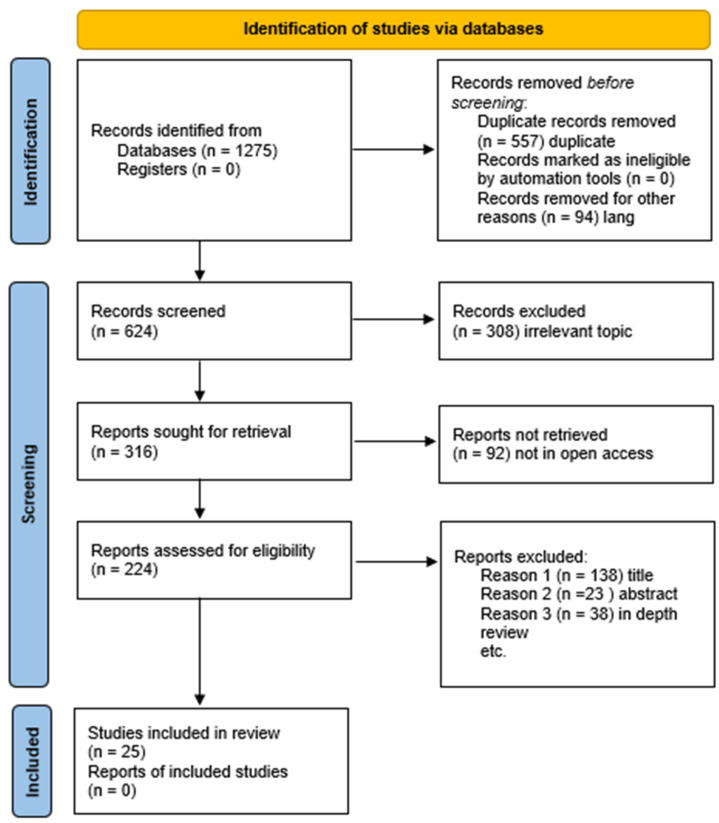
The PRISMA flow diagram depicts the procedure for selecting studies. We conducted a thorough search across multiple globally renowned online databases to identify relevant papers that reported the prevalence of PPR in small ruminants. Subsequently, we employed pre-established search techniques to locate these studies. Once the records were merged and duplicate entries were eliminated, the data underwent a screening process using the pre-established criteria for eligibility before being included in the systematic review and meta-analysis.

**Figure 2 vetsci-11-00280-f002:**
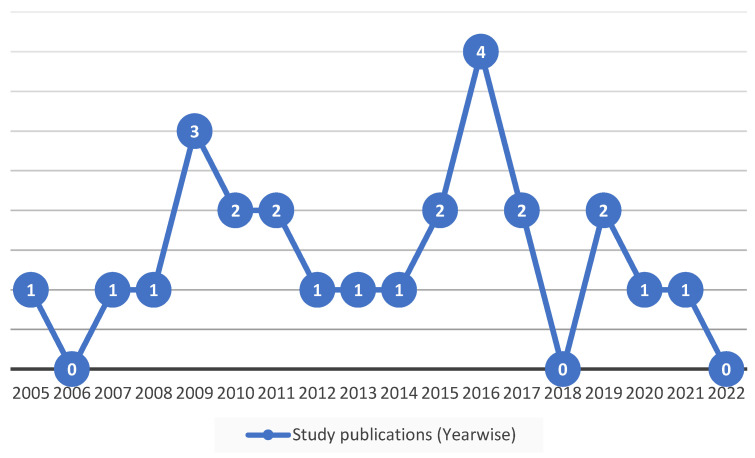
Year-wise published studies spanning the period from 2004–2023. The numeric value shows the number of studies published in the respective year.

**Figure 3 vetsci-11-00280-f003:**
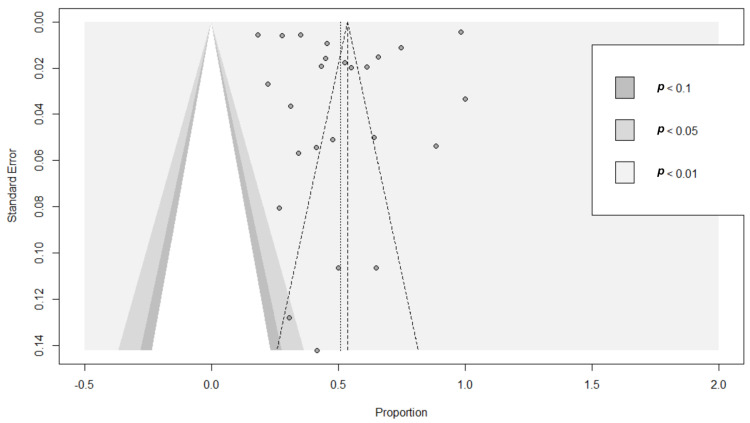
Funnel plot for the analysis of publication bias of the prevalence estimate of PPR in sheep and goats 2004–2023.

**Figure 4 vetsci-11-00280-f004:**
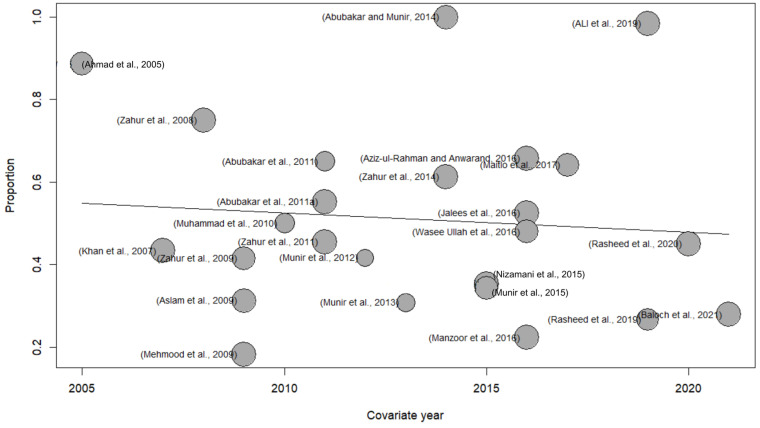
Bubble plot for the univariate meta-regression analysis of PPR in small ruminants [30,31,32,33,34,35,36,37,38,39,40,41,42,43,44,45,46,47,48,49,50,51,52,53,54].

**Figure 5 vetsci-11-00280-f005:**
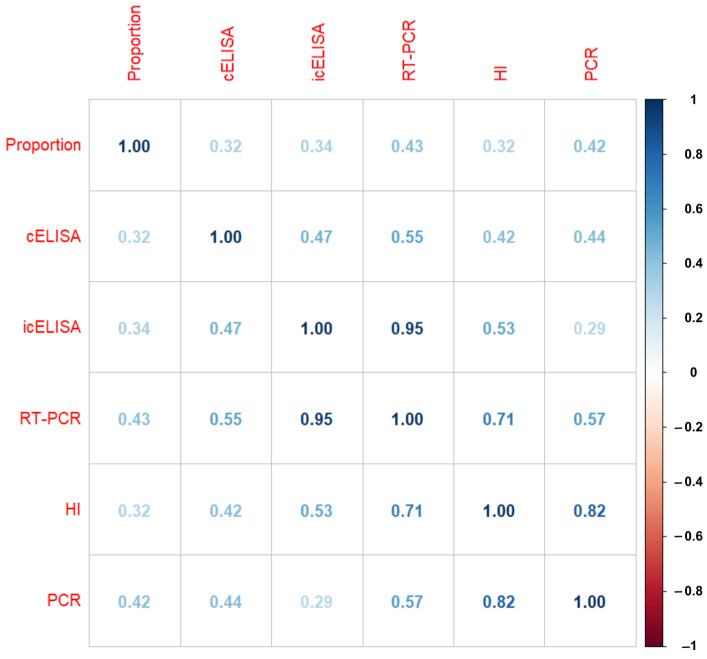
Correlation plot showing the correlation of various detection methods with the proportion of PPR prevalence.

**Figure 6 vetsci-11-00280-f006:**
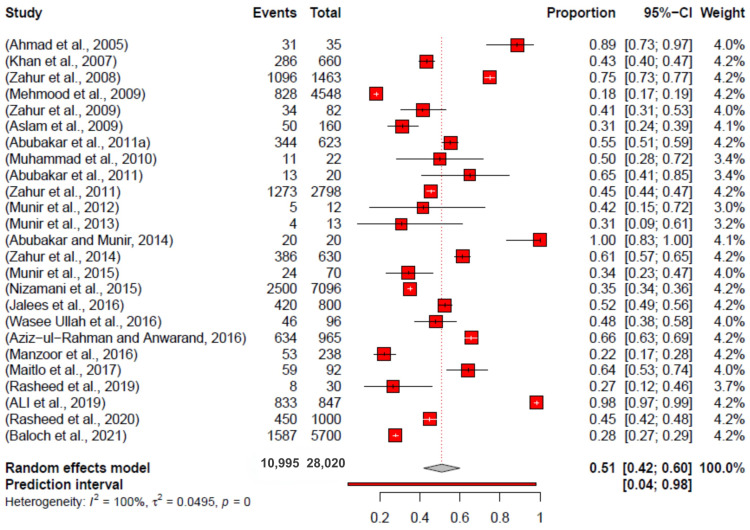
Forest plot of prevalence estimates of PPR in sheep and goats (2004–2023) in Pakistan [30,31,32,33,34,35,36,37,38,39,40,41,42,43,44,45,46,47,48,49,50,51,52,53,54].

**Figure 7 vetsci-11-00280-f007:**
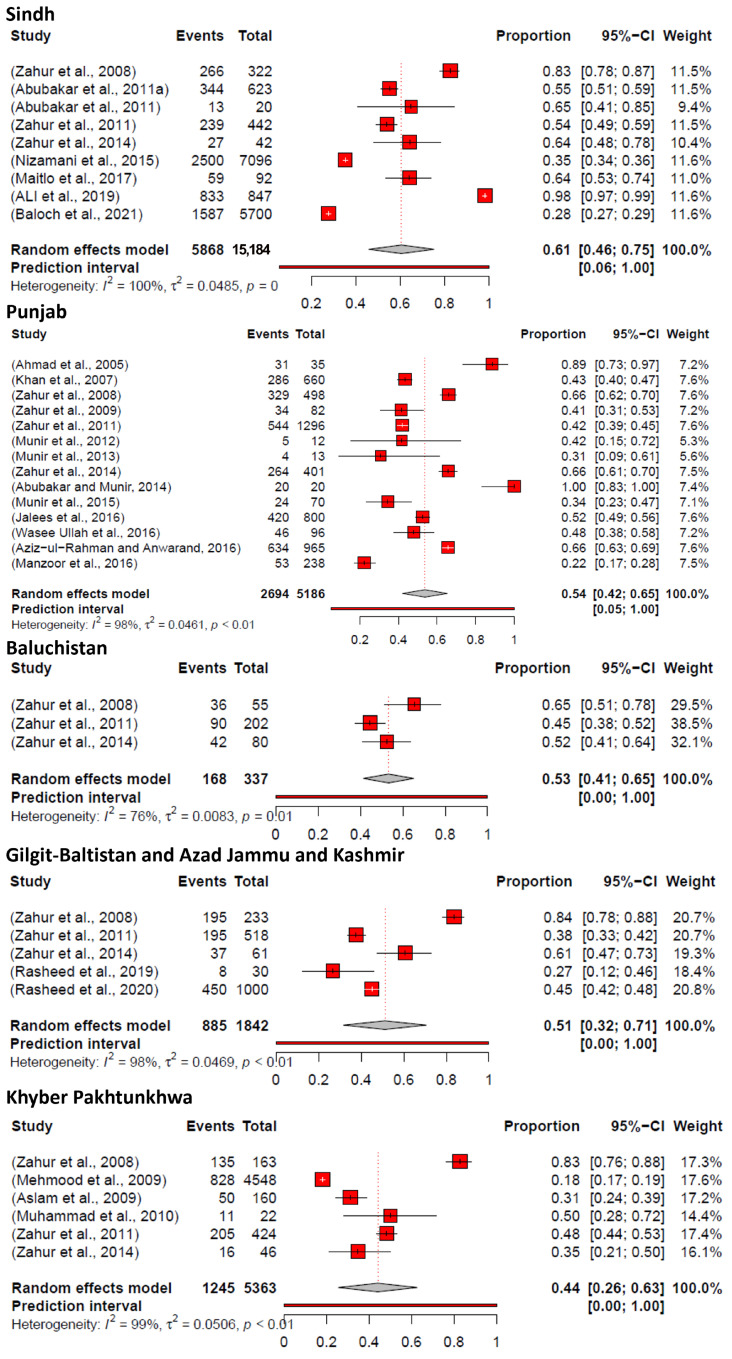
Forest plot for the prevalence estimates of PPR (2004–2023) in Sindh, Punjab, Baluchistan, GB and AJK, and Khyber Pakhtunkhwa. (GB = Gilgit–Baltistan; AJK = Azad Jammu and Kashmir; KPK = Khyber Pakhtunkhwa) [30,31,32,33,34,35,36,37,38,39,40,41,42,43,44,45,46,47,48,49,50,51,52,53,54].

**Figure 8 vetsci-11-00280-f008:**
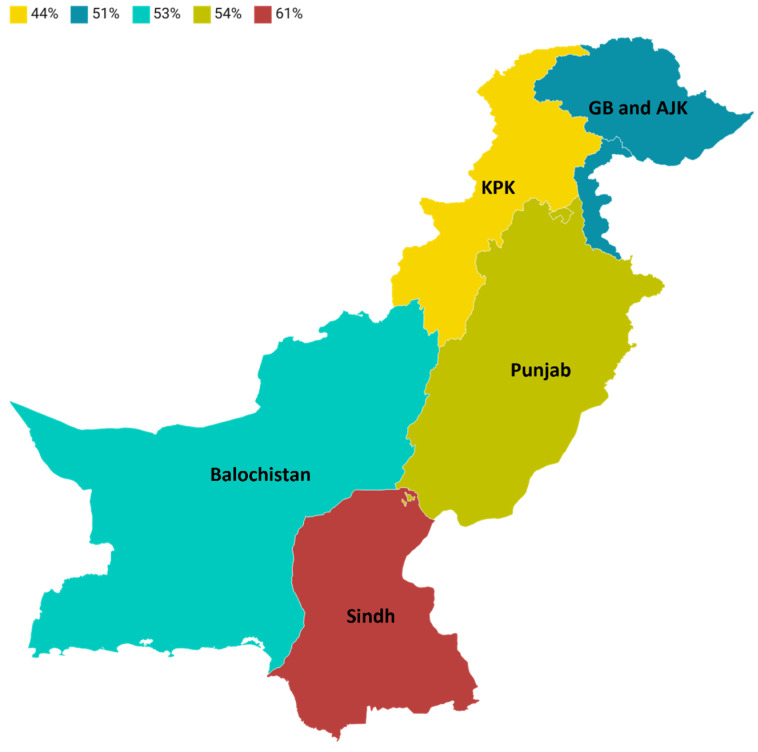
Spatial distribution of prevalence of PPR in domesticated small ruminants in Pakistan. (GB = Gilgit–Baltistan; AJK = Azad Jammu and Kashmir; KPK = Khyber Pakhtunkhwa).

**Figure 9 vetsci-11-00280-f009:**
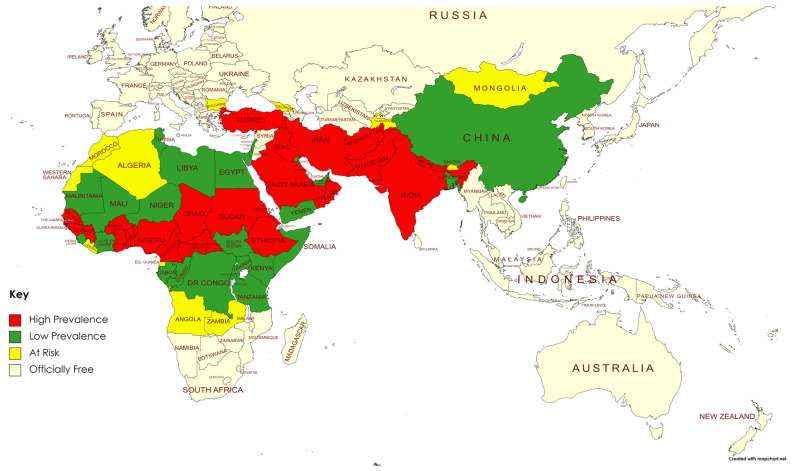
Geographical partition of PPR representing the disease burden.

**Table 1 vetsci-11-00280-t001:** Evidence of PPRV infection in sheep and goats published 2004–2023.

Study Period	Geographical Area	Published Year	Detection Method	No. of Positive Samples	Total Samples	Author
2004–2005	Punjab	2005	c-ELISA	31	35	[30]
2005–2006	Punjab	2007	c-ELISA	286	660	[31]
2002–2005	Pakistan	2008	c-ELISA	1096	1463	[32]
2008	KPK	2009	c-ELISA	828	4548	[33]
2006	Islamabad	2009	Ic-ELISA	34	82	[34]
2008	KPK	2009	c-ELISA, AGID	50	160	[35]
2009–2010	Sindh	2010	c-ELISA	344	623	[36]
2010	KPK	2010	c-ELISA	11	22	[37]
2009	Sindh	2011	Ic-ELISA	13	20	[38]
2005–2006	Pakistan	2011	c-ELISA	1273	2798	[39]
2011	Punjab	2012	Real-Time PCR	5	12	[40]
2012	Pakistan	2013	Real-Time-PCR	4	13	[41]
2005–2007	Pakistan	2014	c-ELISA	386	630	[42]
2011	Pakistan	2015	c-ELISA	24	70	[43]
2012–2013	Sindh	2015	c-ELISA	2500	7096	[44]
2010–2011	Punjab	2016	c-ELISA	420	800	[45]
2012–2013	Punjab	2016	Real Time-PCR	46	96	[46]
2015	Punjab	2016	HI	634	965	[47]
2016	Punjab	2016	c-ELISA	53	238	[48]
2016	Sindh	2017	c-ELISA	59	92	[49]
2005–2006	Pakistan	2017	c-ELISA	5389	19,575	[50]
2017–2018	Gilgit–Baltistan	2019	PCR	8	30	[51]
2016	Sindh	2019	Ic-ELISA	833	847	[52]
2016–2017	Gilgit–Baltistan	2020	HI	450	1000	[53]
2021	Sindh	2021	c-ELISA	1587	5700	[54]

**Table 2 vetsci-11-00280-t002:** Pearson’s chi-squared test showing the level of independence between the prevalence of PPR in sheep and goats.

X-Squared	df	*p*-Value
2.5534	15	0.9999

## Data Availability

Not available due to authors’ reservations.

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
