# Peer review of "Recapitulation of Peste des Petits Ruminants (PPR) Prevalence in Small Ruminant Populations of Pakistan from 2004 to 2023: A Systematic Review and Meta-Analysis"

_vetsci, 2024, doi:10.3390/vetsci11060280_

Round 1
Reviewer 1 Report
Comments and Suggestions for Authors
PPR is an important livestock disease that has both a major economic impact at a country level and survival impact on subsistence farmers.
The authors have performed a literature search and meta-analysis of the data from published papers to determine the prevalence of PPRV in Pakistan between 2004-2023. The search resulted in 25 papers to analyse. The data extraction methods are clearly explained and the analysis appropriate for a prevalence study. However what would enhance the study is a clearer presentation of analysis of different assay methods and prevalence and other covariates that may affect prevalence.
The authors have overstated their conclusions (see my corrections for the Simple Summary and Abstract) as they really only show prevalence rates in different regions. They do not have any analysis on eg stocking density, husbandry practices, or movement or on sheep vs goats to be able to make any conclusions that these are linked to prevalence.
Major edits
Simple summary is not simple – includes some very technical result terms. These should be removed and the conclusion from the numerical results used. However the data show that the values of both groups are within the CI of the other group ie they are not different. They therefore cannot say ‘Specific regions had higher prevalence rates, emphasizing the necessity for focused control actions in those particular areas.’
The study has not analysed different risk factors and so cannot say in the summary line 24 ‘the investigation has found other factors that influence the spread of PPR, including animal husbandry techniques, immunization, and geographical factors.’
Do not use ‘meta-analysis’ (line 26)– can use something like ‘review of the literature’ would be simpler.
Abstract
As above, the prevalence rates are within the CI of each group and so cannot be said to be different (line 39-41).
Results
Figure 3/line 226-229 – the text says that the use of detection methods had a notable impact on the total diversity and yet the Figure only shows year of analysis. Please show the correlation graph for detection method vs Proportion.
Line 237-242 – does the use of a specific test correlate with increasese prevalence – there is no data shown for this.
Reorder Figure 4 so that the reference data for a specific region are together. Add a column after the reference and put the region into the figure. This would allow removal of Figures 4A-E. This may mean that the references in Figure 4 need to be used more than once to show the different regions. The way the authors have used Figure 4 and then split again into Figures 4A-E over separate figures is very strange.
Move Figure 5 to the part of the paper where the metaanalysis methods are shown – lines 205-206
Line 316-318 – when was the data in the published meta-analysis performed.
Line 332, 345-347 – all the prevalence rates are within the CI’s for the separate regions so the authors cannot say that one region has a higher prevalence than another unless they perform some other statistical test. This affects their discussion about different control measures in different regions.
Line 356 – the analysis does not look at risk variables. There has been no analysis of any of the factors listed with prevalence of the virus.
Line 379 – where is the data showing that prevalence was higher in goats than in sheep?
Minor edits
Title line 2-3: ‘Prevalence in Small Ruminants Population’ should be ‘Prevalence in Small Ruminant Populations’
Simple summary
Line 22, 40 – ‘conferred by’ should be ‘seen in’
Line 23 – ‘Specific regions had higher prevalence rates,’ should be ‘Several regions had high prevalence rates,’
Line 39 – change ‘&’ to ‘and’
Line 54 – ‘yet’ should change to ‘however’
Line 55 – ‘Although’ change to ‘In addition,’
Line 60 – ‘(PPR) is sometimes’ change to ‘(PPR), sometimes’
Line 61 – add a comma after ‘’goat plague’ [6]
Line 64 - After first use and definition of ‘Peste des petits ruminants virus (PPRV)’ use only PPRV eg on line 68, similarly use PPR where appropriate
Line 85 – ‘PPR’ should be ‘PPRV’
Line 83-88 – the two sentences are repetitive
Line 89 – use centigrade not Fahrenheit: ‘may’ should be ‘that may’
Line 90 – delete ‘turn out’; ‘anorexia is there and muzzle is dried.’ should be ‘there is anorexia and the muzzle may be dry.’
There are a lot more minor English errors after this that I have not shown that need to be corrected.
Line 92 – ‘short’ should be ‘snort’
Line 107 – ‘underlying’ should be ‘current’
Line 139 – ‘systemic’ should be ‘systematic’
Line 163 - ‘scraped data’ what does this mean?
Line 196 – ‘complied’ should be ‘compiled’
Line 222, 265, 271, 277, 284, 292, 300 – ‘a heterogenicity’ should be ‘heterogeneity’
Line 225-226 – there appears to be a stray sentence – ‘Please provide…..’
Line 232 – ‘weightage’ should be ‘weighting’
Line 262-263, 268-269, 274-275, 281-282, 289-290, 297-298 – remove brackets and delete n = from the sample numbers
Line 309-313 – remove the two sentences starting from ‘Table 1….’ It is repeating the sentences above.
When citing figures or tables in the text, do not use brackets.
Comments on the Quality of English LanguageMultiple minor English problems throughout the paper.
Reviewer 2 Report
Comments and Suggestions for Authors
This is a review of previous papers examining the prevalence pf PPRV in various regions in Pakistan from 2004 to 2023. There is some value in bringing this information together. It would appear that the authors have used correct statistical approaches and have acknowledged limitations of the study. My major concern is they have not discussed the vaccination status of animals. Are all those serologically positive cases due to infection or could some be due to vaccination? The data may be available in the individual papers, and needs discussed.
Minor points.
English language and wording needs improved in a number of places.
1. Line 36. Do the authors mean 'gathered' rather than 'scraped'?
2. Line 55. Should start ‘However’ and not 'Although'
3. Line 60. Remove first 'is' and replace with comma.
4. Line 81 There is not a relative comparison being made in this sentence so the word 'respectively' can't be used. What therefore is the 100% and what is the 50 to 80%?
5. The information in line 83 is repeated again in line 86. Please remove.
6. Line 89 is poorly written 'Should be ' Clinically acute disease is associated with elevated fever (up to 106 deg F which may last for a week), depression, anorexia and dry muzzle'. The authors also need to give a description of the sub-acute form of the disease since they refer to this in in line 88.
7. Line 377. The authors need to explain their remark about use of serum samples. The vaccination status and longevity of morbillivirus antibodies need to be discussed here.
Comments on the Quality of English LanguageOther areas of English editing are required. Reading by a natural English speaker would imporve.
Reviewer 3 Report
Comments and Suggestions for Authors
Dear Authors,
I have reviewed your article titled "Recapitulation of Peste des Petits Ruminants (PPR) Prevalence in Small Ruminants Population of Pakistan during 2004 to 2023: A Systematic Review and Meta-Analysis." Your study provides valuable insights into the epidemiology of PPR in Pakistan, synthesizing data from 25 selected articles out of 1275 initially identified from various databases.
Please see the following recommendations:
Line 154: Please remove "4) Articles that were published before 2004"
Line 225: Please remove "Please provide the outcomes or define the details you want".
Moreover, You calculated an overall pooled prevalence of PPR in Pakistan to be 51% (95% CI: 42-60) with substantial heterogeneity (I² = 100%, 𝜏² = 0.0495, P = 0). I commend you for acknowledging and addressing potential selection bias within your study.
However, I would like to bring to your attention that 24% of the studies within the funnel plot suggest potential publication bias. Nonetheless, your meta-regression analysis indicated non-significant results (P>0.05) for the impact of sample size on effect size, suggesting that publication bias may not substantially influence the overall findings.
Despite these minor limitations, your study significantly contributes to our understanding of PPR prevalence in Pakistan. With minor adjustments to address the aforementioned points, I believe your manuscript is suitable for publication.
Best regards,
Comments on the Quality of English LanguageNo comments to add.
